# Nitro-Oleic Acid Inhibits Stemness Maintenance and Enhances Neural Differentiation of Mouse Embryonic Stem Cells via STAT3 Signaling

**DOI:** 10.3390/ijms22189981

**Published:** 2021-09-15

**Authors:** Jana Pereckova, Michaela Pekarova, Nikoletta Szamecova, Zuzana Hoferova, Kristyna Kamarytova, Martin Falk, Tomas Perecko

**Affiliations:** 1Institute of Biophysics of the Czech Academy of Sciences, Department of Cell Biology and Radiobiology, Kralovopolska 135, 612 65 Brno, Czech Republic; pekarova@ibp.cz (M.P.); szamecova@ibp.cz (N.S.); zhofer@ibp.cz (Z.H.); 460438@mail.muni.cz (K.K.); falk@ibp.cz (M.F.); tomas.perecko@ibp.cz (T.P.); 2Department of Biochemistry, Faculty of Science, Masaryk University, Kamenice 5, 625 00 Brno, Czech Republic

**Keywords:** nitro-oleic acid, pluripotency, mouse embryonic stem cells, STAT3, neurogenesis, cardiomyogenesis

## Abstract

Nitro-oleic acid (NO_2_-OA), pluripotent cell-signaling mediator, was recently described as a modulator of the signal transducer and activator of transcription 3 (STAT3) activity. In our study, we discovered new aspects of NO_2_-OA involvement in the regulation of stem cell pluripotency and differentiation. Murine embryonic stem cells (mESC) or mESC-derived embryoid bodies (EBs) were exposed to NO_2_-OA or oleic acid (OA) for selected time periods. Our results showed that NO_2_-OA but not OA caused the loss of pluripotency of mESC cultivated in leukemia inhibitory factor (LIF) rich medium via the decrease of pluripotency markers (NANOG, sex-determining region Y-box 1 transcription factor (SOX2), and octamer-binding transcription factor 4 (OCT4)). The effects of NO_2_-OA on mESC correlated with reduced phosphorylation of STAT3. Subsequent differentiation led to an increase of the ectodermal marker orthodenticle homolog 2 (*Otx2*). Similarly, treatment of mESC-derived EBs by NO_2_-OA resulted in the up-regulation of both neural markers *Nestin* and β-Tubulin class III (*Tubb3*). Interestingly, the expression of cardiac-specific genes and beating of EBs were significantly decreased. In conclusion, NO_2_-OA is able to modulate pluripotency of mESC via the regulation of STAT3 phosphorylation. Further, it attenuates cardiac differentiation on the one hand, and on the other hand, it directs mESC into neural fate.

## 1. Introduction

Mouse embryonic stem cells (mESC) can be easily propagated in vitro in the presence of external stimuli such as leukemia inhibitory factor (LIF), bone morphogenetic protein 4, and serum [1]. Mouse ESC cell fate is controlled via the tight interplay between cell-intrinsic mechanisms, which are regulated by cell-extrinsic signals [2]. Among others, the signal transducer and activator of transcription 3 (STAT3) are maintaining mESC in an undifferentiated pluripotent state [3,4,5]. In general, STAT3 activation is associated with proliferation, inhibition of apoptosis, and cellular transformation. On the other hand, it could suppress tumor growth and induce differentiation and apoptosis in a specific context [6,7]. It also plays an indispensable role in LIF-mediated mESC self-renewal. LIF treatment results in the activation of the heterodimeric receptor complex consisting of LIF and glycoprotein 130, which further recruits and phosphorylates a number of factors, including STAT3 and subsequently the expression of pluripotent genes (e.g., octamer-binding transcription factor 4 (*Oct4*), (sex-determining region Y)-box 2 (*Sox2*), and *Nanog*) [5,8]. Besides mESC pluripotency, activation of STAT3 is essential in the differentiation process as well. Although *Stat3* gene expression is predominant in tissues of mesodermal origin (e.g., cardiomyocytes), it can directly or indirectly suppress genes, which control mesodermal and endodermal lineage commitment and thus inhibit differentiation towards these germ layers [9]. With regards to cardiomyocytes in vitro, beating embryoid bodies (EBs) are characterized by higher levels of phosphorylated STAT3 compared to non-beating EBs, and *Stat3^-/-^* mESC-derived EBs resulted in lack of the expression of several cardiac-specific genes (e.g., *alpha-actin*, NK2 Homeobox 5 (*Nkx2.5*)) [10]. Moreover, the heart-specific knockout of STAT3 resulted in higher sensitivity to heart failure in juvenile mice [10]. Similar to cardiomyogenesis, the total loss of STAT3 in differentiated EBs is associated with a decrease of neuronal precursor cells and down-regulation of neuronal-specific genes (e.g., *Nestin*) [11]. However, the essential role in this process is given just by the STAT3 phosphorylation at serine 727 [12].

In recent years, there has been an increasing interest in deciphering the mechanisms responsible for the regulation of ESC pluripotency and differentiation, especially due to the possible use of stem cells in the therapy [13,14,15]. Many of these studies were focused on the direct or indirect regulation of STAT3 signaling pathway [9,10]. Among others, nitro-fatty acids, especially nitro-oleic acid (NO_2_-OA), have been recently described as a modulator of STAT3 activity [16]. Moreover, significant involvement of NO_2_-OA in various pathological states was also described [17,18,19].

Nitro-fatty acids (NO_2_-FAs) are naturally occurring electrophiles that are generated via the action of reactive nitrogen species on cellular lipids, mainly polyunsaturated fatty acids [20]. A measurable increase in the levels of NO_2_-FAs has been shown under different pathological conditions such as ischemia reperfusion injury or after bacterial infection [17,18]. Numerous studies have reported that NO_2_-FAs trigger pleiotropic signaling actions including an anti-inflammatory and antioxidant response [18,21,22]. To date, NO_2_-OA has been described to post-translationally modify different protein structures, functions, and/or subcellular protein distribution [16,22].

While the role of fatty acids in the regulation of stem cell pluripotency and differentiation was already studied [2,23], the effects of NO_2_-FAs, especially NO_2_-OA, in this process remain unclear.

## 2. Results

### 2.1. NO_2_-OA Significantly Inhibits Expression of Pluripotency Markers in mESC

To evaluate the role of NO_2_-OA on the maintenance of stemness, the model of mESC in the presence or absence of LIF was employed. First, the potential cytotoxic effect of NO_2_-OA was studied (Appendix A) using experimental setup (Appendix AA). Data showed that NO_2_-OA had no toxic effect on the cells in the concentrations used, as proven by cell cytotoxicity assay (Appendix AB,C). Next, the specific pluripotency-associated genes (*Nanog*, *Sox2*, and *Oct4*) and proteins (NANOG, SOX2, and OCT4) were analyzed (Figure 1) using experimental setup (Figure 1A). As expected, LIF withdrawal caused a significant drop of expression in the observed genes in the control cells (Appendix A) by experimental setup (Appendix AA). The same pattern was detected regardless of the presence of OA or NO_2_-OA. While in the absence of LIF, treatment did not affect the levels of genes compared to the control (Appendix AB–D), the presence of both NO_2_-OA and LIF significantly decreased the expression of all genes (Figure 1B–D). Importantly, OA±LIF co-treatment did not affect mESC pluripotency. Indeed, all obtained results were confirmed at the protein levels (Figure 1E–G; Appendix AE–G).

### 2.2. NO_2_-OA-Dependent STAT3 Signaling Regulation Is Responsible for the Reduction of mESC Stemness

To determine the mechanism of NO_2_-OA action on mESC pluripotency in the presence of LIF, specific signaling pathways essential in mESC stemness were studied (STAT3, ERK, and AKT signalization) (Figure 2) using experimental setup (Figure 2A). Importantly, phosphorylation of STAT3 protein was significantly reduced in NO_2_-OA treated cells. No significant modification was observed in the case of OA treatment (Figure 2B). On the other hand, the treatment of NO_2_-OA did not change the phosphorylation as well as the total level of both ERK and AKT proteins (Figure 2C).

### 2.3. The Differentiation Process Is Changed by NO_2_-OA Treatment

Based on the results showing the reduction in mESC stemness in the presence of LIF, the differentiation process using EBs formation (Figure 3A) was studied. First, the sizes of EBs were analyzed (Figure 3B) as indicated at a representative picture of 5ds EBs (Figure 3C). Neither NO_2_-OA nor OA changed the size of EBs. Then, the gene markers of all germ layers were analyzed at the initial phase of differentiation (at day 5), namely *Otx2*-ectoderm, *Brachyury*-mesoderm, and *Hnf4α*–endoderm (Figure 3D–F). Although both NO_2_-OA and OA treatment changed the expression of *Otx2*, significant up-regulation was observed only in NO_2_-OA treated cells (Figure 3D). Neither the expression of the mesodermal marker nor the expression of the endodermal marker was altered after the treatment with either NO_2_-OA or OA (Figure 3E,F).

Further, the changes in cardiomyogenesis and neurogenesis were evaluated at phenotypical-maturated cells (at day 14 and 15) (Figure 4) using experimental setup (Figure 4A). Importantly, gene expressions of all cardio-specific markers (myosin heavy chain 7 (*Myh7*), myosin light chain 7 (*Myl7*), and cardiac troponin T (*cTnT*)) as well as the percentage of beating EBs were significantly reduced after NO_2_-OA treatment at day 15 (Figure 4B–E). Similar to previous results, OA treatment did not change the levels of these markers compared to the control, and although the percentage of beating EBs was lower from day 10 to day 13, these changes were not significant (Figure 4B–E). On day 14, the percentage of beating EBs treated with OA was similar to the control (Figure 4E). Further, in contrast to NO_2_-OA treated cells, the beating areas in control and OA groups were clearly observed (Appendix A). With respect to neurogenesis, the changes of *Nestin* (neuronal progenitor marker) and *Tubb3* levels (neuron-specific marker) were analyzed. The gene expression of both neuro-specific markers was significantly up-regulated in NO_2_-OA-treated cells compared to both control and OA treatment at day 15 (Figure 4F–G). Neurogenisis was also evaluated using morphology studies, neurites were clearly visible at NO_2_-OA-treated cells at day 14 (Figure 4H; Appendix AA,B) [24,25].

## 3. Discussion

In this study, we have described a previously unknown effect of NO_2_-OA in the regulation of LIF-associated pluripotency in mESC. Specifically, NO_2_-OA significantly downregulates LIF-mediated STAT3 phosphorylation, which is critically involved in the maintenance of stemness in mESC. Our study also addressed the question of NO_2_-OA role in the regulation of mESC differentiation. Importantly, a positive impact on neurogenesis was observed, as demonstrated by increased mRNA levels of *Nestin* and *Tubb3*, and changes in cell morphology.

The role of fatty acids in the regulation of cell stemness was described mostly in cancer stem cells [26,27]. Little is known about the role of fatty acids in the regulation of stem cell pluripotency. The research of Wang et al. provided information about the importance of de novo synthesis of fatty acids as a critical regulator of ESC stemness via affecting mitochondrial fission [2]. However, the role of NO_2_-FA in the process of stem cells stemness has not been studied yet.

It is well known that stem cells self-renewal is controlled via cell-intrinsic mechanisms, which are regulated by cell-extrinsic signals. The cell-intrinsic mechanisms involve the “stemness” factors, OCT4, SOX2, and NANOG, as core regulators of ESC self-renewal [8]. Thus, a decrease of their levels, as was shown in the present study for NO_2_-OA+LIF treated group, was indicative of significantly reduced mESC pluripotency. With regard to extrinsic signals, in serum conditions, LIF/STAT3 are the most important factors for maintaining pluripotency of mESC [13,28]. Here, we showed that NO_2_-OA but not OA suppressed LIF-associated pluripotency of mESC, most probably via the reduction of STAT3 phosphorylation. Besides LIF/STAT3, other pathways were identified as important regulators of ESC fate, including ERK, AKT, and WNT pathways [3,4,14]. However, none of these pathways was affected by NO_2_-OA, which corresponds with the results showing that neither OCT4, SOX2, nor NANOG levels were changed by NO_2_-OA in the conditions without LIF. Similar to our data, the reduction of STAT3 phosphorylation by NO_2_-OA was already reported, however, in that study, the ERK1/2 signaling pathway was affected [16]. This could be explained by engaging different cell line models and different NO_2_-OA concentrations, pointing to possibly cell type-specific response and importance of the experimental conditions.

Besides maintaining pluripotency, STAT3 is important in mESC differentiation as well. Although *Stat3* gene expression is predominant in tissues of mesodermal origin (e.g., cardiomyocytes), it can directly or indirectly suppress genes that control mesodermal and endodermal lineage commitment and thus inhibit differentiation towards these germ layers [9]. With respect to our results, the decrease of STAT3 phosphorylation in mESC via NO_2_-OA treatment did not change the expression of mesoderm or endoderm gene markers in 5 days EBs. However, similar to the research of Foshay et al. [10] showed that inhibition of STAT3 results in the absence of expression of several cardiac-specific genes, we detected a decrease of all *Myh7*, *Myl7*, and *cTnT* after downregulation of NO_2_-OA-mediated STAT3 phosphorylation. Moreover, Foshay et al. also found out that beating areas were characterized by higher levels of STAT3 compared to non-beating areas [10]. In line with this, we were unable to detect beating EBs within 14 days of differentiation in the NO_2_-OA treated group. It is well known that except for cardiomyogenesis, neuronal differentiation originated from the ectoderm is also affected via STAT3 activity. The total loss of STAT3 in differentiated EBs is associated with a decrease of neuronal precursor cells [11]. In contrast to the study of Foshay et al., our data showed that NO_2_-OA treatment of EBs results in a significant increase of neuronal markers *Nestin* and *Tubb3* (selected by [29]), and the presence of neurites was observed (see the morphology study for the details [25]). This discrepancy in the literature could be explained by the fact that NO_2_-OA inhibits the STAT3 phosphorylation at tyrosine 705 residue. However, the loss of STAT phosphorylation at serine 727 residue is crucial for neuronal differentiation [12]. These and our findings thus provide primary insights for future research.

In conclusion, the most significant result emerging from our data is that NO_2_-OA is able to regulate both pluripotency as well as differentiation of mESC. On the one hand, NO_2_-OA significantly reduces pluripotency, which correlates with the regulation of STAT3 phosphorylation. Similarly, it attenuates cardiac differentiation presented by both the downregulation of cardiac-specific genes and the decrease of beating EBs. On the other hand, it increases the levels of neuronal-specific genes and thus promotes neurogenesis resulted in changes in cell morphology.

## 4. Materials and Methods

### 4.1. Chemicals

Oleic acid (OA) (3 M) was purchased from Sigma-Aldrich (O1383; St. Louis, MO, USA). NO_2_-OA (3 M) was kindly provided by Bruce A. Freeman (Department of Pharmacology and Chemical Biology, University of Pittsburgh, Pittsburgh, PA, USA). Both OA and NO_2_-OA were diluted in methanol (100 mM) and stored at −80 °C. Before each experiment, OA and NO_2_-OA were diluted in methanol (10 mM) and further in Dulbecco’s Modified Eagle’s Medium (0.1 mM) (#D4947, DMEM, HyClone, Logan, UT, USA) and immediately used.

### 4.2. Cultivation and Treatment of mESC

The mESC line R1 (ATCC^®^ SCRC-1011™) was cultivated as described previously [15]. Cells were routinely tested for the presence of Mycoplasma contamination (RT-qPCR). Before the treatment, mESC were seeded in the complete growth medium for 24 h, then the medium was replaced to experimental medium, which was a complete growth medium with or without LIF (#LIF2010, Sigma; St. Louis, MO, USA) containing 7.5% of FBS (#10270106, Gibco, Waltham, MA, USA) and mESC were treated with OA or NO_2_-OA (10 µM).

To evaluate the effect of NO_2_-OA on stem cell renewal, the cells were seeded as described above and treated for 72 h with OA or NO_2_-OA (10 µM).

To analyze the phosphorylation of STAT3, extracellular signal-regulated kinase (ERK) and AKT cells were cultivated with the experimental medium with 7.5% FBS for 1 h, further OA or NO_2_-OA (10 µM) were added for 15 min (for more details, see Figure 1).

### 4.3. Differentiation of mESC

To determine the role of NO_2_-OA treatment in mESC on the cell-differentiation status, the cells were seeded in the complete growth medium for 24 h. Then, the medium was replaced with an experimental medium with LIF containing 7.5% of FBS, and mESC were treated with OA or NO_2_-OA (10 µM). After 72 h, suspension of mESC (2.5 × 10^6^ cells/mL) was directly seeded on the top of AggreWellTM400 dish (#34415, StemCell Technologies; Vancouver, BC, Canada) into the complete medium with LIF and 15% FBS to form uniform EBs. After 24 h of incubation (day 0), the EBs were gently transferred onto a 1% agarose-coated dish and cultivated in a complete medium without LIF. On day 5, the EBs were harvested and analyzed.

To analyze the size of embryonic bodies (EBs), the EBs were prepared as described above. At day 0, the individual EBs (in triplicate) were seeded on an agarose-coated 24-well dish. The size of EBs was determined at day 1 and day 5 using a light microscope, Dino-Eye Edge Eyepiece camera, and DinoCapture v.2.0 software (Dino-Lite Digital microscope; Taipei, Taiwan).

To evaluate the effect of NO_2_-OA on the differentiated progenitors, the cells were differentiated as described above. Briefly, at day 0, EBs (formed from the mESC suspension) were transferred onto 1% agarose-coated dish and cultivated in a complete medium without LIF. On day 5, the EBs were seeded on gelatin-coated dishes into DMEM/F-12 (1:1) medium (#21331020, Gibco; Carlsbad, CA, USA) supplemented with insulin-transferrin-selenium (#41400-045, Gibco; Carlsbad, CA, USA) and 100 IU/mL penicillin and 0.1 mg/mL streptomycin (Sigma; St. Louis, MO, USA) for further 10 days. From day 6 to day 10, the cells were treated with OA or NO_2_-OA (10 µM) (for more details, see Figure 1).

These time points represent various stages of cell development: progenitors (up to day 5) and phenotypical-maturated cells (around day 15).

For counting the beating EBs, the EBs were prepared as described above. At day 0, the individual EBs were seeded on an agarose-coated 24-well dish. From day 6 to day 10, the cells were treated with OA or NO_2_-OA (10 µM), and from day 10 to day 14, the beating of EBs was observed using a light microscope.

### 4.4. RNA Isolation and Quantitative Real-Time PCR (qRT-PCR)

Mouse ESC cells and differentiated cells were washed with PBS and lysed using the HighPure RNA Isolation Kit (Roche; Basel, Switzerland). The concentration and purity of isolated RNA was assessed using a Spectrophotometer Infinite M200 Pro (Tecan, Grödig, Austria). Only samples with an A280/A260 absorbance ratio greater than 1.9 were used for further investigation. Total RNA (1 μg) was reversely transcribed into first-strand cDNA using the Transcriptor First Strand cDNA Synthesis Kit according to the manufacturer’s protocol (#04379012001, Roche; Basel, Switzerland). qRT-PCR reactions were performed in a LightCycler480 instrument using LightCycler480^®^ Probes Master solutions (Roche; Basel, Switzerland), and the following program was used: an initial denaturation step at 95 °C for 10 min, followed by 45 cycles (95 °C for 10 s, 60 °C for 30 s, and 72 °C for 1 s), and a final cooling step at 40 °C for 1 min. The sequences of primers and numbers of Universal Probe Library (UPL) probes are listed in Table 1. Data were normalized to ribosomal protein L13A (Rpl13a) and presented as 2^−Δcq^.

### 4.5. Protein Expression Analysis by Western Blot Technique

Total protein lysates were prepared from undifferentiated mESC and differentiated cells using RIPA Lysis and Extraction Buffer (cat. no. 89901) containing Protease and Phosphatase Inhibitor (cat. no. 78442) (both from Thermo Scientific; Waltham, MA, USA). Proteins were quantified using the BCA Protein Assay Kit (Cat. No. 23227, Thermo Fisher Scientific, Waltham, MA, USA). A total of 20 µg of protein lysates were loaded on SDS-polyacrylamide gel. Immunoblot analyses were performed as presented previously [15]. The following primary antibodies were used to detect particular proteins: rabbit anti-OCT-3/4 (#9081, 1:1000) from Santa Cruz Biotechnology (Dallas, TX, USA), rabbit anti-NANOG (#8822, 1:1000), rabbit anti-SOX2 (#23064, 1:1000), rabbit anti-phospho-STAT3 (#9145, Tyr705, 1:2000), mouse anti-STAT3 (#9139, 1:1000), anti-total/phospho-ERK1/2 (#4696, 1:2000/#4377, 1:1000), anti-total/phospho-AKT (#2920, 1:2000/#4056, 1:1000) (all from Cell Signaling Technology, Danvers, MA, USA), mouse anti-Vinculin (V9264, 1:10000; Sigma-Aldrich, St. Louis, MO, USA). Corresponding secondary HRP-conjugated anti-rabbit or anti-mouse antibodies were employed. Luminescence signal was read on LAS-3000 Imaging System (FujiFilm, Tokyo, Japan) (Figure 1) and/or Amersham Imager 680 (GE Healthcare, North Richland Hills, TX, USA) (Figure 2). Relative levels of proteins were quantified by scanning densitometry using the ImageJ™ program v1.48 (National Institutes of Health, Bethesda, MD, USA).

### 4.6. Data Analysis

All results were calculated as a percentage of non-treated control except the number of beating EBs (Figure 4E) were the percentage of beating EBs in each well, which is presented in the graph. Data are presented as mean ± standard deviation of at least 3 independent measurements. Representative images of Western blot analysis of phospho- and total-AKT and phospho- and total-ERK2 (Figure 2C) express results obtained from 2 (AKT), respectively, 3 (ERK) independent experiments.

All data passed the Shapiro–Wilk test for normal distribution. The effect of NO_2_-OA was compared to i) non-treated control to show the non-affected processes and to ii) OA-treated samples to prove the specific effects of NO_2_^−^ group. Results were statistically analyzed using an unpaired student’s *t*-test, two-tailed (GraphPad Prism v.9.1.0., GraphPad Software, USA). All significant outlier values were removed based on Grubbs’ test, also called the extreme studentized deviate (GraphPad Prism v.9.1.0., GraphPad Software, USA). Statistical differences compared to non-treated control were expressed in the graph. A * *p*-value of less than 0.05 was considered significant.

## Data Availability

The data presented in this study are available on request from the corresponding author.

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
