# Peer review of "Nitro-Oleic Acid Inhibits Stemness Maintenance and Enhances Neural Differentiation of Mouse Embryonic Stem Cells via STAT3 Signaling"

_ijms, 2021, doi:10.3390/ijms22189981_

Round 1

Reviewer 1 Report

The paper entitled "Nitro-oleic acid inhibits stemness maintenance and enhances neural differentiation of mouse embryonic stem cells via STAT3 signaling" describes the NO2-OA involvement in the regulation of stem cell pluripotency and differentiation.

My main concerns are about Paragraph 2.3 The differentiation process is changed by NO2-OA treatment 

First, the authors do not analyse proportion of EBs but size (Fig 3B), so please correct the text. A proper quantification concerning EB sizes should be added to Fig 3B close to the representative  pictures. So the relative qualification of 5d EBs should be moved from fig S3 to fig 3. Fig S3 is not mentioned in the main text. Regarding fig S3 I'm wondering if every dot in the graph refers to a single EB and if they come from a single batch. If it is the case, the number of analysed EBs has to be increased to give significance to the result. 

 In order to claim the activation of germ layers markers upon NO2-OA treatment, and in particular the ectoderm, more the a single marker is required. The authors should clarify also the passage from the germ layers markers analysis to the cardiomyogenesis and neurogenesis and explain which is the origin of this tissues and why they are interested in these tissues in particular on Cardiomyogenesis. The authors show the beating areas in FigS4 B and C, indicated by circles. How is it possible to define them as beating areas? A movie should be included or a specific  immunostaining for beating areas like STAT3 as suggested by Reference 10 or appropriate markers.

The authors refer to Nestin as a neuro-specific marker, but without any reference. A more appropriate neuronal staining instead of Nestin, that is mainly a neuronal progenitor marker, is required, like DCX or TUJ1. The authors should also explain how do they define neurites only based on morphology Fig 4G. Moreover another marker for neuronal progenitors should be added to better claim that NO2-OA treatment is interfering with the effect of STAT3 in EBs differentiation as mentioned in the discussion.

The assessment in the Abstract : "The effects of NO2-OA on mESC and differentiated EBs were mediated via  reduced phosphorylation of STAT3" should be changed since the authors do not show reduced phosphorylation of STAT3 on differentiated EBs . 

Minor points:

Figure 1 and 2: These figures refer only to cultures +LIF. Please correct the scheme in 1A and 2A.

Fig1D: One more experiment point should preferably be added in the treatment +NO-OA since one point seems a clear outlier.

Fig2C: Quantifications relative to the representative WB should be added.

Fig S5 and S6 cited in the main text are missing in the supplemental figures!

Reviewer 2 Report

Dear Members of the Editorial Team:

Re: ijms-1182043-peer-review-v1:  “Nitro-oleic acid inhibits stemness maintenance and enhances 2 neural differentiation of mouse embryonic stem cells via 3 STAT3 signaling. “

The work presented investigates the recently discovered transductor, NO2-OA.  The authors demonstrated that applying NO2-OA to murine embryonic stem cells (mesc) in the presence of LIF leads to the loss of pluripotent markers.  These effects were not achieved with OA.  Furthermore, differentiation in the presence of NO2-OA upregulated the ectoderm linage (ectoderm Otx2 upregulated).  Upon the formation of embryonic bodies in the presence of NO2-OA, the neural phenotype is shown by the upregulation of neural-specific nestin.  By contrast, cardiac differentiation was decreased.  Together, STAT3 phosphorylation was reduced with NO2-OA. 

The paper will be acceptable with revisions.

1) In Figure 2B, both the western blot and graph of pSTAT3 for phosphorylation of NO2-OA treatment do not appear significant between OA and NO2-OA.  What is the reduction of NO2-OA compared to OA?

2) In Figure 3C, what was the upregulation of Otx 2 caused by NO2-OA compared to OA.  The increase resulting from NO2-OA appears not to be substantially greater relative to OA.

Round 2

Reviewer 1 Report

Few comments:

1_ please correct +LIF only in the legend of Figure 2 

2_Unfortunately the format of the movies in Fig S4 and S5 is not accessible for my Apple computer so I cannot watch them.

Anyway I still find not clear the beating areas in FigS3 B and C, indicated by circles. The figure legend states: "Beating areas of control (B) and OA treated (C) cells were visualized at day 14 of differentiation using light microscope". But these pictures are in my opinion not informative without an appropriate IHC. Maybe better to leave the video only.

3_The authors write in the cover letter:

The neurites were define based on i) our previous results (Vecera et al. (2017) [PMID 29422917]) where the first author of this manuscript (Pereckova) was one of the co-authors (previous last name Kudova), and on ii) the results of Sexana et al. (2020) [PMID 33083240] showing the differentiation of mESC towards neural cells, where the similarities of morphology are obvious.

Since it is not obvious for me that mESC differentiated for 14d in culture are all  neuroblasts/neurons and then the cell processes can be only neurites (and not processes from neural progenitors), I suggest to add the proper reference or to refer to previous work when describing neurites in Fig 4H without IHC.

Author Response

This manuscript is a resubmission of an earlier submission. The following is a list of the peer review reports and author responses from that submission.